# Strategies for Modifying Adenoviral Vectors for Gene Therapy

**DOI:** 10.3390/ijms252212461

**Published:** 2024-11-20

**Authors:** Anna Muravyeva, Svetlana Smirnikhina

**Affiliations:** Laboratory of Genome Editing, Research Centre for Medical Genetics, Moskvorechye, 1, 115522 Moscow, Russia

**Keywords:** adenoviral vectors, gene therapy, immunogenicity, tropism, tissue-specific promoters, capsid modifications, inherited diseases, cystic fibrosis, ornithine transcarbamylase deficiency

## Abstract

Adenoviral vectors (AdVs) are effective vectors for gene therapy due to their broad tropism, large capacity, and high transduction efficiency, making them widely used as oncolytic vectors and for creating vector-based vaccines. This review also considers the application of adenoviral vectors in oncolytic virotherapy and gene therapy for inherited diseases, analyzing strategies to enhance their efficacy and specificity. However, despite significant progress in this field, the use of adenoviral vectors is limited by their high immunogenicity, low specificity to certain cell types, and limited duration of transgene expression. Various strategies and technologies aimed at improving the characteristics of adenoviral vectors are being developed to overcome these limitations. Significant attention is being paid to the creation of tissue-specific promoters, which allow for the controlled expression of transgenes, as well as capsid modifications that enhance tropism to target cells, which also play a key role in reducing immunogenicity and increasing the efficiency of gene delivery. This review focuses on modern approaches to adenoviral vector modifications made to enhance their effectiveness in gene therapy, analyzing the current achievements, challenges, and prospects for applying these technologies in clinical practice, as well as identifying future research directions necessary for successful clinical implementation.

## 1. Introduction

Adenoviruses are viral particles measuring 70–100 nm, with an icosahedral capsid and no outer envelope. Their genome consists of linear double-stranded DNA ranging from 25,000 to 45,000 base pairs in length (Figure 1). Adenoviral vectors are genetically modified adenoviruses in which part of the DNA has been replaced with a transgene. These vectors are often based on human adenovirus type 5, which uses coxsackievirus and adenovirus receptors (CARs) to enter cells. Vectors based on other serotypes may use different cellular receptors, allowing targeted action on specific cell types and increasing transduction efficiency [1]. Currently, AdVs are actively used in gene therapy for the treatment of infectious diseases, cancer, and vaccine development [2].

However, adenoviral vectors have several drawbacks that limit their effectiveness and safety in clinical applications. One of the main drawbacks is low specificity, which leads to insufficient transduction efficiency of target cells. This means that the virus can infect not only the intended cells but also others, reducing therapeutic efficacy and increasing the risk of side effects [3]. Additionally, adenoviruses have high immunogenicity, eliciting a strong immune response in the patient. This leads to the development of side effects and limits the possibility of repeated vector administration, as the immune system quickly recognizes and neutralizes the reintroduced virus [4].

On the other hand, the immunogenicity of adenoviral vectors can be considered an advantage in the context of immunotherapy, as they promote a secondary immune response against tumor cells. Thus, this property of adenoviruses enables their use in cancer therapy [5]. This approach has been utilized in the development of approved adenoviral drugs for cancer treatment, such as Gendicine and Oncorine, which are used in clinical practice in China. Gendicine, approved in 2003, is an adenoviral vector expressing the *p53* gene, which stimulates apoptosis in tumor cells. This drug was developed for the treatment of head and neck cancer [6]. The immunogenicity of Gendicine enhances its antitumor effect by recruiting immune cells to the tumor site and facilitating the destruction of cancer cells. Another drug, Oncorine, approved in 2006, is also adenovirus-based and used for head and neck cancer therapy. The immune response to the adenoviral vector contributes to the destruction of tumor cells by activating the immune system [7].

However, despite the successes in using adenoviral vectors for cancer therapy, no drugs based on adenoviruses have been developed for gene therapy for hereditary diseases to date. This is due to the significant limitations of adenoviral vectors, which make it challenging to use them safely and effectively for treating hereditary diseases, where sustained expression of the therapeutic gene and minimization of side effects are necessary. Nevertheless, it is worth noting that researchers are actively working on modifying adenoviral vectors to improve their specificity and effectiveness. One approach involves the use of tissue-specific promoters to control transgene expression, allowing the vector’s action to be directed toward specific cell types and reducing the likelihood of affecting non-target cells. There is also ongoing research on modifying the AdV capsid to enhance tropism to target cells, which increases the likelihood of achieving a therapeutic effect. Capsid modifications can also help reduce the immunogenicity of the vectors, thereby minimizing side effects after dose administration (Figure 2). These strategies aim to create safer and more effective adenoviral vectors for successful use in clinical practice.

To enhance clarity and provide a structured overview of the primary strategies covered, a summary table is included below (Table 1). This table outlines the high-level strategies discussed in the article, detailing each approach’s rationale, brief descriptions categorized by subsections, and relevant references. The strategies encompass genetic modifications, capsid modifications, approaches to reducing immunogenicity, and techniques for modifying tropism to improve specificity and efficacy in adenoviral vector-based gene therapy. This structured format aims to facilitate a comprehensive understanding of each section’s key points and their implications for clinical and research applications.

## 2. Tissue-Specific Promoters for Controlling Transgene Expression

Using tissue-specific promoters in adenoviral vector delivery restricts transgene expression to cells where the promoter is active, minimizing cellular toxicity and immune clearance. The challenge of controlling transgene expression duration and levels arises from the lack of specific promoters and regulatory elements in the adenoviral genome. Optimal promoters can regulate transgene expression more effectively.

Scientific advances in the development of adenoviral vectors using cell- or tissue-specific promoters are primarily aimed at cancer gene therapy, as AdVs have demonstrated the ability to effectively deliver therapeutic genes to tumor cells [8]. Cancer cell-specific promoters can regulate the expression of tumor suppressor genes, tumor-specific antigens, and other immunomodulatory molecules, enhancing the efficacy of gene therapy and reducing side effects. For example, in one study, researchers created a dual tumor-specific vector system using PEGylation and a telomerase reverse transcriptase (TERT) promoter for tumor therapy through systemic vector administration in mice. This approach resulted in higher tumor-selective transgene expression compared to the administration of unmodified AdV [9].

In another study, researchers developed conditionally replicating adenoviruses with a cyclooxygenase (Cox-2) tissue-specific promoter to achieve promoter-controlled expression of the *E1* gene for viral replication and increased transduction efficiency of pancreatic cancer cells in mouse models. The intratumoral administration of such adenoviral vectors with the Cox-2 promoter in mice showed a stronger antitumor effect and led to an increase in the number of viral copies in the tumor [10].

To enhance the efficacy of cystic fibrosis gene therapy, an AdV with a *lacZ* gene encoding beta-galactosidase was created under the control of an airway-specific promoter consisting of a 2 kbp 5’-untranslated region of the *CFTR* gene. The study revealed that this vector directs beta-galactosidase expression in cell lines expressing *CFTR* and in human and mouse airway cells in vitro and in vivo, indicating successful transgene expression. However, the study noted the lack of strict specificity of *CFTR* gene expression in airway cells compared to endogenous *CFTR*. This suggests that, despite the presence of airway-specific promoters in the AdV, such a vector does not express the *CFTR* transgene predominantly in the targeted cell populations for which its specificity was enhanced by the incorporated promoter [11].

Another study aimed to increase *CFTR* gene expression levels in airway cells by creating an adenoviral vector with a strong promoter specific to these cells. These vectors contained the k18 promoter, an epithelial cell expression cassette of keratin 18, which regulates the expression of reporter genes β-galactosidase (*β-gal*) or human alpha-fetoprotein (*AFP*). The constructed AdV was tested in mouse models through intranasal administration. The results demonstrated high levels of reporter gene expression in epithelial and submucosal airway cells. Moreover, the administered vector dose was sufficient for effective cell transduction, with no significant side effects, and the level of transgene expression was maintained for a longer duration. The results suggest the potential of this vector for use in cystic fibrosis gene therapy [12]. However, despite promising results, a drug based on this approach has not yet been developed, and further studies are needed to optimize safety and efficacy before clinical application.

Tissue-specific promoters have also been used in studies of gene therapy for OTCD (Ornithine Transcarbamylase Deficiency). It has been shown that an adenoviral vector expressing the human alpha1-antitrypsin (*hAAT*) gene, controlled by liver-specific promoters, did not induce neutralizing antibodies in all mice. Thus, the choice of promoter and, accordingly, tissue-specific regulation of gene expression can affect the host immune response, allowing for the circumvention of the humoral immune response [13].

Despite the potential of new developments to improve adenoviral vectors and in vivo testing, these advancements have not reached clinical trials. This is likely due to ongoing challenges with transgene expression control and other issues limiting their use in gene therapy for inherited diseases.

## 3. Capsid Modifications to Change Tropism

Achieving therapeutic gene expression requires high vector doses due to low transduction efficiency in target cells. Adenoviral vectors’ broad tropism allows them to enter many cell types but does not guarantee sufficient expression in target cells. High doses can cause dose-dependent toxicity and an immune response, limiting gene transfer efficacy and risking severe side effects. Increasing AdV specificity through capsid modifications can improve targeting to specific cell types.

### 3.1. Capsid Pseudotyping

Pseudotyping involves replacing capsid proteins responsible for binding to cell receptors with proteins from other virus strains or even from different viruses with alternative tissue tropisms to alter the specificity of cell infection. For instance, it has been demonstrated that replacing the capsid proteins of adenovirus serotype 5 with capsid proteins from Ad35 can enhance transduction efficiency in CD34 hematopoietic cells [14].

Another method of capsid pseudotyping involves modifying the fiber knob protein, which plays a key role in binding the adenoviral vector to cell receptors. Recent studies have investigated changes to this protein to increase tropism and improve the targeting efficiency of adenoviral vectors. For example, it has been shown that the chimeric Ad5/49K vector, created by replacing the fiber knob of adenovirus serotype 5 with that of serotype 49, more effectively penetrates dendritic and vascular smooth muscle cells by binding to the CAR receptor [15]. These results confirm the potential of the chimeric Ad5/49K as a vector for effective vascular gene therapy, as well as for use in vaccine development, highlighting its promise for clinical application.

Different human adenoviruses contain arginine–glycine–aspartic acid (RGD) sequences, which allow them to interact with integrins that bind vitronectin (αv). Adding the RGD sequence to the capsid proteins of adenoviral vectors, such as the fiber knob, enhances their ability to bind to integrins on the surface of cells, especially cancer cells, thus increasing the virus’s efficiency in cell entry and its selectivity for tumor cells. The strategy of adding the RGD sequence to adenovirus fibers was tested in a Phase I clinical trial for the treatment of recurrent glioblastoma. Patients received a serotype 5 adenovirus with a modified fiber knob in the capsid, which included the RGD sequence [16]. This modification allowed the adenovirus to selectively bind to integrins on the surface of tumor cells, ensuring optimal efficiency in cell entry and infection. Additionally, the deletion of 24 base pairs in the adenoviral *E1A* transcription gene prevented replication in healthy cells. The study demonstrated the safe administration of the adenoviral vector into the brain, localized inflammation, and an improvement in the condition of some subjects.

Through pseudotyping, an adenoviral vector potentially useful in gene therapy for cystic fibrosis was created. This vector is based on adenovirus serotype 5 (Ad5) and contains capsid proteins from Ad35, which use CD46 receptors located on the apical membrane of respiratory epithelial cells for cell entry. Thus, the pseudotyped AdV5 shifted its tropism from CAR receptors located on the basolateral membrane, which is challenging to access due to tight epithelial junctions between airway cells, to more accessible receptors on the apical surface of the cells [17].

One study found that adenovirus serotype 17 more effectively transduced respiratory epithelial cells in cystic fibrosis compared to Ad5 and Ad2, which are most commonly used as the basis for gene therapy vectors for cystic fibrosis [18]. The increased efficiency of cell infection is likely due to the composition of Ad17 capsid proteins. Researchers created a vector based on adenovirus serotype 2 with capsid proteins from Ad17. Improved targeting to airway epithelial cells and increased efficiency of transgene expression were demonstrated in vitro. Another example is Ad5F37, a pseudotyped Ad5 with capsid proteins from Ad37 that uses sialic acid, rather than CARs, as a receptor for cell entry [19].

A key limitation of pseudotyping is the limited availability of viral envelope proteins for targeting specific cell receptors. Effective targeting requires knowledge of receptor expressions on target cells and their exclusivity to specific cell types. While pseudotyping enhances AdV tropism, it can cause off-target transgene expression in cells with similar receptors. To address this, capsid modifications with adapter molecules are used.

### 3.2. Capsid Conjugation with Molecular Adapters

Conjugating adenoviral capsids with molecular adapters, like bispecific antibodies, allows for the targeting of specific cell receptors by binding one end to the target cell and the other to a capsid attachment protein. This approach redirects AdVs to the desired cells, removing native tropism and creating new tropism toward target cells. A key advantage is that different adapters can bind to the same vector without requiring genetic modifications, making this method simple and versatile for various studies.

Most research is aimed at conjugating AdVs with molecular adapters to enhance the effect of gene therapy for cancer, as many different pathways are regulated during the tumor process [20]. For example, in one study, single-chain variable fragments (scFv) were fused with epidermal growth factor (EGF) proteins to enhance the tropism of AdVs to cells expressing EGF receptors, thereby increasing the effectiveness of gene therapy [21]. Another example is an adapter that links the end of a vector’s capsid to a biotin-acceptor peptide that undergoes biotinylation in cells, which can be used to alter AdV tropism through conjugation with biotinylated antibodies [22]. Another approach was the use of polyethylene glycol as an adapter linking AdVs to fibroblast growth factor 2 (FGF2) to improve the tropism of vectors to ovarian cancer cells [23]. An additional example of capsid conjugation with molecular adapters is PeptiCRAd technology, which is an oncolytic adenovirus conjugated with tumor-specific peptides [24]. This approach provides selectivity of adenoviruses for tumor cells and enhances immunogenicity, leading to a stronger immune response against cancer cells. The effectiveness of this technology has been demonstrated in various in vivo studies for different types of tumors [25,26,27,28].

Notable studies in cystic fibrosis gene therapy have used adapters as target ligands to enhance the tropism of AdVs to respiratory epithelial cells. After discovering the urokinase plasminogen activator receptor on the apical surface of the human airway epithelium, adenoviral vector targeting was performed on these cells using a peptide adapter linked to the AdV capsid via polyethylene glycol. The results showed a significant increase in the transduction efficiency of epithelial cells in vitro [29].

A variation of capsid conjugation with molecular adapters is the formation of receptor–ligand complexes. The idea is that the normal ability of the viral capsid to bind its receptors is altered or eliminated, and instead, a small peptide ligand is incorporated into the capsid structure to bind to alternative receptors expressed on the target cell. For example, a mutation in the adenovirus fiber gene, which typically binds to CARs, was altered to bind integrin instead. This increased gene transfer to cells that normally lack the CAR receptor, allowing the virus to successfully infect previously inaccessible cells [30]. Another study found that removing the ability of adenovirus to bind CARs was sufficient to alter its targeting to brain cells [31]. Another example is forming a complex between the CAR receptor of the adenoviral vector and a ligand binding the CD40 receptor expressed by dendritic cells. These vectors demonstrated the high transduction efficiency of dendritic cells [32]. In another study, a complex was formed between the same CAR capsid protein and a single-chain antibody against human carcinoembryonic antigen, targeting AdVs to colorectal cancer metastases in the liver [33]. It is worth noting that the above studies were conducted in vivo.

Another approach to enhancing the tropism of adenoviral vectors to specific cell types is creating libraries of alternative viral capsid variants using the phage display method, significantly improving the transduction efficiency of target cells [34]. This method is based on selecting peptides from a phage library that can bind to target cell receptors, forming a complex with the viral protein responsible for cell attachment, ensuring high transduction efficiency. In this process, peptides that bind to both the viral capsid and receptors on the surface of target cells are used.

The advantage of this approach compared to genetic modifications of AdVs is that there is no limitation on the size of the insertion that can be incorporated into the adenoviral vector. Additionally, the absence of genetic engineering manipulations helps maintain vector stability and does not adversely affect its assembly process in target cells [35].

In one study, to target the adenoviral vector to C2C12 mouse skeletal muscle cells, a phage library of 12 amino acid random peptides was created, inserted between the H and I sheet of the AdV fiber protein, which were incorporated into the pIII protein of the fd bacteriophage. As a result of the selection, a peptide was chosen and translated into the fiber protein that binds much better to C2C12, leading to a significant increase in cell transduction efficiency with the modified adenoviral vector [36].

## 4. Reducing Immunogenicity

Adenoviral vectors have high immunogenicity, leading to rapid clearance of transduced cells and a reduction in therapeutic gene expression duration, causing immune reactions and side effects. For gene editing, tumor therapy, or vaccination, repeated AdV administration may not be needed. However, gene replacement therapy for inherited diseases requires sustained transgene expression. Additionally, pre-existing antibodies against common adenovirus serotypes and AdV proteins’ high immunogenicity limit repeated vector use in the same patient.

### 4.1. Anti-Inflammatories

The immunogenicity of adenoviral vectors can be reduced with anti-inflammatory drugs. For example, dexamethasone has been shown to reduce the inflammatory response to the administration of an adenoviral vector, thereby lowering immune side effects that may limit the effectiveness of gene therapy. When dexamethasone was administered to hemophilic mice receiving an adenoviral vector with the *FVIII* gene, an increased expression of the *FVIII* gene was observed compared to the group without dexamethasone. There was also a significant reduction in the level of the pro-inflammatory cytokine TNF-alpha and the liver enzyme alanine aminotransferase, indicating reduced liver toxicity [37].

The use of various anti-inflammatory drugs, such as dexamethasone, paracetamol, diclofenac, ibuprofen, and ketorolac, was analyzed to understand their effect on adenovirus entry into cells, which is crucial for improving the efficiency of gene therapy with adenoviral vectors. The introduction of these drugs can impact the immunogenicity of vectors and their interaction with cells, allowing for a reduced inflammatory response and enhanced viral entry into target cells, thereby increasing gene delivery efficiency. The study was conducted in vitro on SiHa cell cultures and in vivo on BALB-C mice. The results showed that dexamethasone, paracetamol, and ibuprofen enhanced adenovirus entry into both SiHa cells and mouse liver tissues, with the most significant changes observed in the liver. This suggests the potential to reduce immune barriers, which in turn improves gene expression levels and reduces the likelihood of adverse side effects. Diclofenac increased viral entry only in vitro in SiHa cells, while ketorolac had no significant effect [38].

Other research evaluated the effect of inflammation suppression with dexamethasone on the immune response and the duration of transgene expression in adenovirus-mediated transfection in the nasal mucosa of mice. For this purpose, the recombinant adenovirus Ax1CAlacZ, containing the *Escherichia coli* beta-galactosidase gene (*lacZ* gene), was introduced into the nasal mucosa of mice that were pre-treated with dexamethasone or not treated with it. The results showed that dexamethasone significantly increased mRNA levels on days 4, 7, and 14 and prolonged the expression of the beta-galactosidase protein compared to the control group, where expression had almost disappeared by days 7 and 14. This study demonstrates that inflammation suppression with dexamethasone reduces the immune response and supports prolonged transgene expression, making this approach promising in terms of enhancing the effectiveness of adenoviral gene therapy [39].

Thus, the use of anti-inflammatory drugs such as dexamethasone may be a promising strategy to improve the effectiveness of adenoviral gene therapy by modulating the cellular environment and immune response, facilitating increased viral entry and reducing vector immunogenicity.

### 4.2. Chemical Capsid Modifications

#### 4.2.1. Modifications by Synthetic Polymers

A major advantage of chemically modifying adenoviral vectors with polymers is that it can be done after production and purification, allowing for thousands of modifications on the capsid surface. However, a drawback is that this process must be repeated after each vector assembly and purification, as these modifications are not encoded in the genetic material and do not appear in newly produced vectors [40].

Polyethylene glycol (PEG) is a synthetic, non-charged polymer with high hydrophilicity, low immunogenicity, and low toxicity. PEGylation, or the chemical modification of protein preparations based on the covalent attachment of PEG, is often used in research because it reduces immunogenicity, increases solubility, and positively affects biological activity in vivo [41,42]. Therapeutic proteins used in clinical practice include PEG-alpha-interferon, PEG-adenosine deaminase, and PEG-interleukin 2 [43,44].

When modifying adenoviral capsids with PEG, the thiol groups of cysteine residues or the amino groups of lysine residues are bound. It has been shown that a large number of amino groups are present on the capsid surface of human adenovirus serotype 5, making them the most common targets for PEG [45].

For example, in one study, a PEGylated adenovirus (PEG-Ad) was created. It demonstrated resistance to neutralization by antibodies even in the presence of high titers of antibodies against adenovirus, indicating that PEGylation effectively protects the vector from the host immune system. This suggests that PEGylation may allow repeated administration of AdVs without a loss of efficacy due to immune memory. The efficiency of gene expression after PEGylation was evaluated using A549 cells. The results showed that PEG-AdV could still effectively deliver the transgene to target cells. Moreover, an increase in the plasma half-life of the vectors was observed, suggesting prolonged systemic circulation time in the body and, consequently, potentially improved transduction efficiency [46].

A study investigated the effect of PEGylation of adenoviral vectors on reducing the immune response in mice to improve the efficiency of cystic fibrosis gene therapy. PEGylated first-generation AdV, when administered intratracheally to mice, exhibited low immunogenicity, as evidenced by a decrease in cytotoxic T-lymphocytes and a significant increase in the duration of transgene expression. Additionally, low concentrations of neutralizing antibodies against the AdV capsid were noted [47]. In another study, following the intranasal administration of PEG-AdV to mice, no increase in neutralizing antibody titers in the lungs was detected [45].

Poly(N-(2-hydroxypropyl) methacrylamide), or N-HPMA, is another synthetic hydrophilic polymer with low toxicity and immunogenicity, used for modifying adenoviral capsids. It was found that an N-HPMA-modified adenoviral vector was resistant to antibody neutralization [48].

It is known that the use of adenoviral vectors for gene therapy for cystic fibrosis is limited by the formation of neutralizing antibodies to AdVs. A recent study on the administration of an αCD20 antibody as an immunomodulatory agent prior to a dose of an AdV showed that αCD20 significantly reduces the B-cell immune response and immunoglobulin production in the lungs of mice, allowing for safer repeated vector administration and thereby increasing the effectiveness of cystic fibrosis gene therapy [49].

#### 4.2.2. Capsid Pseudotyping

Pseudotyping technology is used to enhance the tropism of adenoviral vectors to specific cell types, but it can also reduce vector immunogenicity. In one study, a vector based on adenovirus serotype 35 was pseudotyped with a fiber knob from adenovirus serotype 5, called Ad35k5. In vitro transduction revealed that the vector used Ad5 CAR receptors for cell entry instead of Ad35 CD46 receptors. In vivo studies in mice and rhesus macaques showed a strong immune response to Ad35k5 compared to control Ad35. Therefore, this study concluded that the fiber knob of Ad5 has high immunogenicity, which can be reduced by modifying Ad5 vectors. Differences in the protein composition of Ad5 and Ad35 capsids explain the variations in the immunogenicity of these viruses [50].

Capsid proteins from animal viruses can also be used in pseudotyping technology. For example, a vector based on human adenovirus serotype 5 was pseudotyped with chimeric fibers of bovine adenovirus serotype 4, HAdV-5-F2/BAdV-4. Intravenous administration of the vector to mice resulted in reduced humoral and innate immune responses. The levels of various cytokines decreased with HAdV-5-F2/BAdV-4 administration compared to unmodified HAdV-5, and the chimeric vector evaded neutralizing antibodies. Therefore, HAdV-5-F2/BAdV-4 is characterized by lower hepatotoxicity and immunogenicity [51].

### 4.3. Genetic Modifications

Helper-dependent adenoviral vectors (HD-Ad) are modified adenoviral vectors devoid of all viral genes, containing only the necessary sequence for the expression of a therapeutic gene. These vectors rely on a helper virus for replication and assembly, significantly reducing immunogenicity and increasing safety for use in gene therapy. Some studies using these vectors in vivo on mouse and primate models have demonstrated a reduced immune response and prolonged transgene expression. When comparing HD-Ad and first-generation AdV vectors containing a leptin transgene in leptin-deficient mice, the former vector showed the best transduction efficiency, lower immunogenicity, and longer transgene expression, leading to increased blood leptin levels and subsequent weight loss in animals [52]. In another study, a recombinant HD-Ad with an alpha1-antitrypsin transgene, AdSTK109, was administered intravenously to mice. They demonstrated prolonged and stable transgene expression for more than 10 months, allowing tissue-specific transcriptional regulation of gene expression to be identified. Thus, higher vector doses can be used to achieve physiological levels of alpha1-antitrypsin without hepatotoxicity [53,54]. Subsequently, the same vector, AdSTK109, and a first-generation AdV were administered to baboons, which showed prolonged and stable transgene expression for over a year, while the first-generation AdV showed gene expression duration from 3 to 5 months. Lower immunogenicity was also observed for AdSTK109 compared to the second vector, which elicited a strong humoral immune response in baboons [55]. Similar studies using helper-dependent adenoviral vectors have been conducted for gene therapy for inherited diseases, such as hemophilia A and B in dogs [56,57], Crigler–Najjar syndrome in rats [58], and genetic deficiencies in mice [59].

It is worth noting that some studies have found a similar immune response to helper-dependent AdVs as to first-generation AdVs. For example, similar levels of T-cell production were observed in HD-Ad-transduced dendritic cells, indicating similar immunogenicity of these vectors [60]. Following intravenous administration of HD-Ad encoding *LacZ* (HD-AdLacZ) and mouse secreted alkaline phosphatase (HD-AdSEAP) to DBA/2 mice, elevated levels of inflammatory cytokines and chemokines in the liver, including IP-10, MIP-2, and TNFα, were observed, like those seen with native first-generation Ad administration. Furthermore, an increase in CD11b-positive leukocytes in the liver was detected within hours of the administration of both HD-AdLacZ and HD-AdSEAP [61].

In the development of gene therapy for cystic fibrosis, HD-Ad vectors are of significant interest due to their ability to ensure long-term expression of the therapeutic gene with minimal immune response, thanks to the removal of viral genes. A recent study explores the use of HD-Ad combined with the CRISPR-Cas9 system for gene therapy targeting cystic fibrosis caused by mutations in the *CFTR* gene. With their high capacity and low immunogenicity, HD-Ad vectors show potential for the effective delivery of CRISPR/Cas9 to respiratory epithelial cells [62]. Another study examined approaches to reducing the immune response to HD-Ad to extend the duration of *CFTR* gene expression in the airways of mice, which is especially important for treating cystic fibrosis. The authors described methods of suppressing immune reactions, specifically investigating the effects of cyclophosphamide on immunomodulation and enhancing *CFTR* expression with repeated HD-Ad delivery to the lungs of mice. Cyclophosphamide significantly reduced T- and B-cell infiltration and the level of anti-HD-Ad antibodies, allowing for sustained *CFTR* expression. Thus, the use of HD-Ad-*CFTR* combined with temporary immunosuppression shows promise for gene therapy for cystic fibrosis [63].

## 5. Mucolytics for Effective Gene Delivery in Cystic Fibrosis

This section will not consider the method of modification of adenovirus vectors, but instead possible strategies that can be applied to improve the efficiency of adenovirus vector transduction when delivering genes to respiratory epithelial cells for gene therapy for cystic fibrosis. In cystic fibrosis, mucus on the surface of the respiratory tract is a dense complex of mucoglycoproteins, which creates a significant problem for the penetration of adenovirus vectors and effective gene delivery [64,65]. It has been shown that mucolytic agents break down components of airway mucus. Temporarily disrupting epithelial tight junctions can increase transduction efficiency and reduce the dose of vector required to achieve a therapeutic effect. Studies have demonstrated the application of L-α-lysophosphatidylcholine (LPC) [66], EGTA [67], polidocanol [68], sodium caprate [69], EDTA [70], calcium phosphate co-precipitation [71], and polycation [72].

LPC is one of the most extensively studied mucolytics and is widely used to enhance the effectiveness of gene therapy. For example, it has been employed in gene therapy for cystic fibrosis using helper-dependent adenoviral vectors in rabbit lungs [66], mice [73], baboons [74], and pigs [75,76,77]. These studies demonstrate significant improvement in the transduction efficiency of respiratory epithelial cells and increased expression levels of the CFTR gene through LPC. Despite the successful results of in vivo gene therapy in animal models, LPC is still not used in clinical trials. This is likely due to the fact that LPC does not achieve a consistent effect in increasing AdV transduction efficiency and requires specific concentration and administration timing for each patient.

When sodium caprate was used as a mucolytic agent alongside adenoviral vectors, a decrease in AdV stability and functionality was observed. For this reason, it was decided to administer AdV separately from sodium caprate [78]. In vivo studies on respiratory epithelial cells in mice found no toxic effects following AdV administration [69]. However, this mucolytic is not widely used in preclinical studies of cystic fibrosis gene therapy using AdV, compared to the more commonly used LPC. This is likely due to the need for two separate administrations of the vector and sodium caprate, whereas LPC can be administered simultaneously with the vector, simplifying the experimental procedure.

EGTA, a calcium chelator, is also used to enhance cystic fibrosis gene therapy efficacy. Intranasal administration of AdV into the lungs of mice following EGTA treatment to disrupt tight junctions showed that the lungs were not inflamed and were indistinguishable from the control group receiving saline [67]. The use of EGTA improved the efficiency of transduction of respiratory epithelial cells in vivo in both mice [67] and rabbits [70]. However, another study revealed a toxic effect of EGTA in gene delivery to mouse airway cells [69]. Furthermore, EGTA is not the most effective mucolytic agent for cystic fibrosis gene therapy. It was found that sodium caprate mediated higher transduction efficiency of adenoviral vectors in lung cells of mice compared to EGTA [78].

EDTA, which has a similar chemical structure to EGTA and also acts as a calcium chelator, was used in a Phase I clinical trial for the treatment of patients with cystic fibrosis who had lung infections caused by Pseudomonas aeruginosa [79]. In this study, EDTA showed no toxic effects or adverse reactions. Moreover, EDTA is already used in intravenous chelation therapy for lead poisoning [80]. However, it was found that EDTA disrupted epithelial tight junctions less effectively than EGTA, which likely led to its exclusion from further clinical trials for cystic fibrosis gene therapy.

Polidocanol (PDOC) has also been employed as a mucolytic agent in some studies. For instance, the efficiency of adenoviral vector transduction of mouse respiratory epithelial cells in vivo was significantly increased with the use of PDOC at concentrations of 0.1–1%, due to its ability to enhance epithelial cell permeability, allowing better vector penetration into target cells [68]. However, compared to LPC, PDOC is not as widely used for improving gene delivery to lung cells. For example, PDOC is more frequently used to damage the surface epithelium of the respiratory tract to facilitate the engraftment of exogenous stem cells administered intratracheally in vivo [81].

## 6. Conclusions

Adenoviral vectors, due to their advantages such as high packaging capacity and broad tropism to various cell types, hold significant potential for gene therapy for inherited diseases. However, their application is complicated by several challenges, including strong immune responses, non-specific tropism, which impairs the vector’s targeting to target cells, and short transgene expression duration. This review has discussed various strategies and technologies developed to overcome these barriers, including the use of tissue-specific promoters, capsid modifications, pseudotyping, and the integration of molecular adapters. Additionally, the development of helper-dependent vectors and the use of mucolytics to improve transgene delivery to target cells represent promising approaches to enhancing the safety and efficacy of adenoviral vectors.

Future research should continue to focus on optimizing these technologies. Studies should be centered on improving tissue-specific targeting to enhance transduction efficiency in target cells and reducing immunogenicity through advanced genetic modifications and chemical methods.

Although preclinical studies have yielded promising results, the authors of this review emphasize the need for further development of various approaches and strategies for modifying adenoviral vectors for their subsequent use in clinical practice to evaluate their efficacy and safety in humans. Successfully modified adenoviral vectors could revolutionize gene therapy for inherited diseases and take their place among viral vectors actively used in clinical practice.

## Figures and Tables

**Figure 1 ijms-25-12461-f001:**
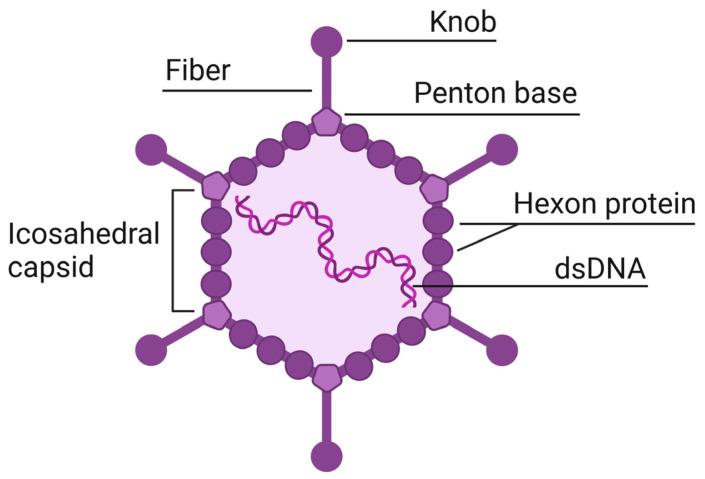
Structure of adenovirus.

**Figure 2 ijms-25-12461-f002:**
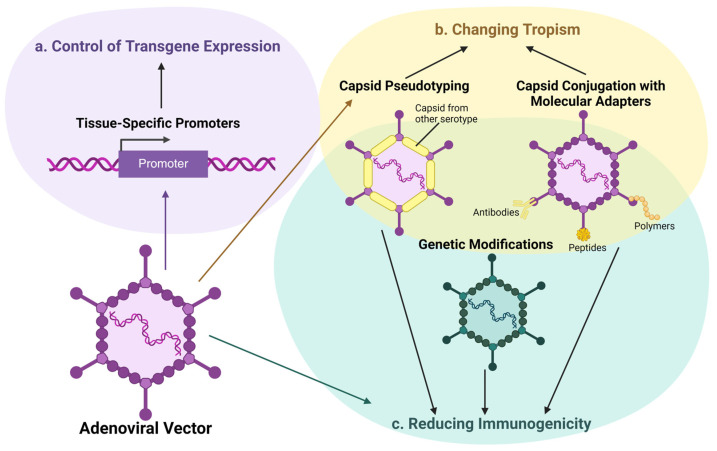
Strategies for modifying adenoviral vectors. (**a**) Control of transgene expression: use of tissue-specific promoters to confine transgene expression to target cells, enhancing safety and minimizing off-target effects. (**b**) Changing tropism: capsid pseudotyping, which replaces capsid proteins with those from other adenovirus serotypes, and conjugation with molecular adapters, which binds capsid proteins to specific cell receptors, are used to alter vector specificity and improve the targeting of cell types, thereby enhancing therapeutic efficacy. (**c**) Reducing immunogenicity: chemical modifications (e.g., PEGylation) or genetic alterations to the capsid to lower immune recognition, allowing for safer repeated administration and prolonged therapeutic effects.

**Table 1 ijms-25-12461-t001:** Summary of key strategies for adenoviral vector-based gene therapy.

High-Level Strategy	Summarized Rationale	Brief Description	Relevant References
Tissue-Specific Promoters	Controlling transgene expression	Promoters targeted specific tissues to control the location of transgene expression and improve precision in therapy.	[8,9,10,11,12,13]
Capsid Modifications	Change tropism to improve targeting	*Section 3.1. Capsid Pseudotyping*: Altering the fiber knob and other capsid proteins to target specific receptors on host cells.	[14,15,16,17,18,19]
		*Section 3.2. Capsid Conjugation with Molecular Adapters*: Conjugating molecular adapters to the capsid surface to improve cell-specific targeting and enhance selectivity.	[20,21,22,23,24,25,26,27,28,29,30,31,32,33,34,35,36]
Reducing Immunogenicity	Reduce immune response to prolong gene expression	*Section 4.1. Anti-Inflammatories*: Utilizing anti-inflammatory agents, such as dexamethasone, to decrease inflammatory responses and improve vector tolerance.	[37,38,39]
		*Section 4.2. Chemical Capsid Modifications and Capsid Pseudotyping*: Applying synthetic polymers or chemical modifications to reduce immune recognition of the capsid.	[40,41,42,43,44,45,46,47,48,49,50,51]
		*Section 4.3. Genetic Modifications*: Genetic alterations to capsid structures to reduce immune detection and increase durability of transgene expression.	[52,53,54,55,56,57,58,59,60,61,62,63]
Mucolytics for Effective Gene Delivery in Cystic Fibrosis	Enhance vector penetration in cystic fibrosis patients	Using mucolytic agents to break down mucus barriers in cystic fibrosis, thereby enhancing vector efficiency for gene delivery to respiratory epithelial cells in cystic fibrosis therapy.	[64,65,66,67,68,69,70,71,72,73,74,75,76,77,78,79,80,81]

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
