# Peer review of "Strategies for Modifying Adenoviral Vectors for Gene Therapy"

_ijms, 2024, doi:10.3390/ijms252212461_

Round 1

Reviewer 1 Report

Comments and Suggestions for Authors

Peer Review Summary

The manuscript presents strategies for modifying adenoviral vectors to tackle challenges related to high immunogenicity, low specificity, and the efficiency of gene delivery. Overall, the manuscript is well-organized and clearly articulated. However, I recommend its acceptance only after addressing the following major concerns:

Major Concerns:

  1. The logic and primary objective of this review paper are clear, as the author discusses the challenges and corresponding strategies to overcome them. However, the background section is somewhat limited. I suggest the authors expand this section to include a more comprehensive overview of the current state of the art and practices in the field.
  2. I recommend adding tables to summarize the key points from Sections 2 to 5. The tables should include high-level strategies, summarized rationales, brief descriptions, and relevant references. This will facilitate quick extraction of the main ideas presented in each section for readers.
  3. Consider including genetic modifications of the fiber protein to enhance specificity. Modifying the fiber knob region of adenoviruses can improve tropism for specific cells or decrease binding to non-target cells.

Minor Concerns:

  • In Line 136, the word “use” should be corrected to “uses.”

Author Response

Dear Reviewer,

Thank you very much for your valuable comments. We have reviewed and addressed each of your suggestions, making revisions to the manuscript accordingly. For your convenience, we have added notes in the document to help you easily locate the corrected sections.

Since Microsoft Word does not allow for different colors to distinguish between reviewers’ comments, we have marked each note at the beginning with the numbers 1, 2, or 3, corresponding to Reviewer 1, Reviewer 2, and Reviewer 3, as indicated in the Editorial Manager. Your review is designated as number 1 (Reviewer 1), so please refer to the comments that begin with the number 1.

Once again, thank you for your insightful feedback, which has greatly helped us improve the quality of our manuscript.

Best regards,
Anna Muravyeva

Reviewer 2 Report

Comments and Suggestions for Authors

The manuscript is interesting, but could be improved with the following suggestions

Please add reference in the following text: This means that the virus can infect not only the intended cells but also others, reducing therapeutic efficacy and in-creasing the risk of side effects. Additionally, adenoviruses have high immunogenicity, eliciting a strong immune response in the patient. This leads to the development of side effects and limits the possibility of repeated vector administration, as the immune system quickly recognizes and neutralizes the reintroduced virus.

Additionally, it is desirable to include the following sections:

1.- Adenoviral vectors approved for commercial use and what modifications they have (or if they lack them). Mentioning their clinical utility and in which countries they were approved.

2.- Somewhere they should mention that immunogenicity can also be an advantage when it comes to immunotherapy, since it has been mentioned that its use can trigger a secondary immune response to cancer, when injected intratumorally.

3.- Possible long-term adverse effects. It has been mentioned that adenoviral vectors can trigger effects by increasing obesity and metabolic diseases in the long term, an aspect confirmed experimentally, but which should be considered and mentioned, since not only short-term effects are possible, but also long-term effects (add reference:

https://pubmed.ncbi.nlm.nih.gov/30666458/

https://pubmed.ncbi.nlm.nih.gov/35621112/

4.- Possible modification of adenovirus tropism by medications, such as anti-inflammatories:

https://www.mdpi.com/2036-7481/15/3/105

https://www.sciencedirect.com/science/article/pii/S0006497119804573

5.- Implications of mass vaccination with adenoviral vectors in COVID (vaccines for covid with vectors adenovirales)

Many of these topics are not the main focus of the manuscript, but briefly adding these aspects will give a clearer idea of ​​adenoviral vectors in general.

Author Response

Dear Reviewer,

Thank you very much for your valuable comments. We have reviewed and addressed each of your suggestions, making revisions to the manuscript accordingly. For your convenience, we have added notes in the document to help you easily locate the corrected sections.

Since Microsoft Word does not allow for different colors to distinguish between reviewers’ comments, we have marked each note at the beginning with the numbers 1, 2, or 3, corresponding to Reviewer 1, Reviewer 2, and Reviewer 3, as indicated in the Editorial Manager. Your review is designated as number 2 (Reviewer 2), so please refer to the comments that begin with the number 2.

Once again, thank you for your insightful feedback, which has greatly helped us improve the quality of our manuscript.

Best regards,
Anna Muravyeva

Reviewer 3 Report

Comments and Suggestions for Authors

The authors summarize approaches to modifying adenoviral vectors to enhance efficacy in gene therapy and the prospects for applying these technologies in clinical settings. The authors' coherent description of adenoviral vector modifications, including tissue-specific expression of transgenes, improved tropism, and reduced immunogenicity, will help advance understanding of the principles of gene therapy using adenoviral vectors. However, there are several points that need to be improved

The authors stated in abstract that “Adenoviral vectors are effective vectors for gene therapy due to their broad tropism, large capacity, and high transduction efficiency, making them widely used as oncolytic vectors, and for crating vectors-based vaccines”. However, it is unclear whether this paper adequately covers adenovirus-based oncolytic virotherapy as well as gene therapy for inherited disease such as cystic fibrosis. This point should be clearly stated in the introduction.

Abstract, lines 16-20: Two sentences with similar content are duplicated. Needs to be corrected.

Figure 2: A more detailed explanation for “a” through “c” is needed in the figure legend. Why not “changing tropism” instead of “enhancing tropism”?

A major point of this paper is the paucity of recent papers in the list of cited papers. Most of the techniques presented were developed around the year 2000, but there should have been some progress in this area by now. More recent research should also be included in the references for a comprehensive review of current advances.

Examples of recently published papers,

Section 3. Capsid modifications to enhance tropism

Bliss CM, et al. A pseudotyped adenovirus serotype 5 vector with serotype 49 fiber knob is an effective vector for vaccine and gene therapy applications. Mol Ther Methods Clin Dev. 2024 Jul 30;32(3):101308. doi: 10.1016/j.omtm.2024.101308.

Nemerow GR. Integrin-targeting strategies for adenovirus gene therapy. Viruses. 2024 May 13;16(5):770. doi: 10.3390/v16050770.

Section 4. Reducing the immunogenicity

Bandara RA, et al. Potential of helper-dependent adenoviral vectors in CRISPR-cas9-mediated lung gene therapy. Cell Biosci. 2021 Jul 23;11(1):145. doi: 10.1186/s13578-021-00662-w.

Cao H, et al. Overcoming immunological challenges to helper-dependent adenoviral vector-mediated long-term CFTR expression in mouse airways. Genes (Basel). 2020 May 18;11(5):565. doi: 10.3390/genes11050565.

Saruuldalai E, et al. Adenovirus expressing nc886, an anti-interferon and anti-apoptotic non-coding RNA, is an improved gene delivery vector. Mol Ther Nucleic Acids. 2024 Jul 16;35(3):102270. doi: 10.1016/j.omtn.2024.102270.

Clark RDE, et al. Evaluation of anti-vector immune response to adnovirus-mediated lung gene therapy and modulation by alphaCD20. Mol Ther Methods Clin Dev. 2024 Jun 24;32(3):101286. doi: 10.1016/j.omtm.2024.101286.

Page 2, lines 61-63: The authors stated that “Scientific advances in the development of adenoviral vectors using cell- or tissue-specific promoters are primarily aimed at cancer gene therapy, as AdV has demonstrated the ability to efficiently deliver therapeutic genes to tumor cells [3]”. Does this paper include the context of “modification of adenoviral vectors to enhance oncolytic capability of vectors”? If so, the following papers should be added to the text and references as recent studies.

Garcia-Moure M, et al. Oncolytic adenoviruses and immunopeptidomics: a convenient marriage. Mol Oncol. 2024 Apr;18(4):781-784. doi: 10.1002/1878-0261.13648.

Chiaro J, et al. Development of mesothelioma-specific oncolytic immunotherapy enabled by immunopeptidomics of murine and human mesothelioma tumors. Nat Commun. 2023 Nov 3;14(1):7056. doi: 10.1038/s41467-023-42668-7.

Koizumi N, et a. Utilizing adenovirus knob proteins as carriers in cancer gene therapy amidst the presence of anti-knob antibodies. Int J Mol Sci. 2024 Oct 3;25(19):10679. doi: 10.3390/ijms251910679.

Page 5, lines 169-170: coxsackie-adenovirus receptor (CAR) should be CAR.

Page 8, lines 335-337: The reference (Hillman et al.) needs to be added to the reference list and the reference number needs to appear in the text.

Comments on the Quality of English Language

none

Author Response

Dear Reviewer,

Thank you very much for your valuable comments. We have reviewed and addressed each of your suggestions, making revisions to the manuscript accordingly. For your convenience, we have added notes in the document to help you easily locate the corrected sections.

Since Microsoft Word does not allow for different colors to distinguish between reviewers’ comments, we have marked each note at the beginning with the numbers 1, 2, or 3, corresponding to Reviewer 1, Reviewer 2, and Reviewer 3, as indicated in the Editorial Manager. Your review is designated as number 3 (Reviewer 3), so please refer to the comments that begin with the number 3.

Once again, thank you for your insightful feedback, which has greatly helped us improve the quality of our manuscript.

Best regards,
Anna Muravyeva

Round 2

Reviewer 1 Report

Comments and Suggestions for Authors

The team has addressed my comments.

Reviewer 2 Report

Comments and Suggestions for Authors

The authors have made the necessary changes to improve the manuscript. Therefore, the manuscript can be published.

Reviewer 3 Report

Comments and Suggestions for Authors

The authors respond appropriately to the points raised by the reviewer.